# Monitoring Subaquatic Vegetation Using Sentinel-2 Imagery in Gallocanta Lake (Aragón, Spain)

**Juan Soria \***[ID]**, Miriam Ruiz and Samuel Morales**

Cavanilles Institute of Biodiversity and Evolutive Biology, University of Valencia, 46980 Paterna, Spain; miruizli@alumni.uv.es (M.R.); samofer@alumni.uv.es (S.M.)
\* Correspondence: juan.soria@uv.es

**Abstract:** Remote sensing allows the study of aquatic vegetation cover in shallow lakes from the different spectral responses of the water as the vegetation grows from the bottom toward the surface. In the case of Gallocanta Lake, its seasonality and shallow depth (less than 2 m) allow us to appreciate the variations in the aquatic vegetation with the apparent color. Six common vegetation indices were tested, and the one with the best response was the so-called NDI45, which uses the normalized ratio between the far red (705 nm) and red (665 nm) bands. Our aims are to show the variations in the surface area covered by vegetation at the bottom of the lagoon, its growth and disappearance when drying occurs, and recolonization in a flooding period. The degree of cover reaches 90% at the most favorable times of the year, generally in summer and coinciding with flooding of the lake. The studied period shows how this method can be used for lacustrine habitat detection and highlights the need for field vegetation inventories in future works, which will allow the spectral measurements to be related to the species present.

**Keywords:** saline wetland; remote sensing; aquatic vegetation; vegetation index; Sentinel-2

## 1. Introduction

Wetlands have significant value for humans and nature: they favor biodiversity and participate in the regulation of the hydrological cycle, among other functions [1]. Their conditions are strongly linked to groundwater, so they are vulnerable to actions such as alterations in or contamination of aquifers, agricultural intensification, and urban expansion [2].

Protected areas such as wetlands have international designations in the Ramsar Convention, which promotes the formation of an international network of wetlands and thus the conservation of those wetlands included in the Ramsar List [3]. A total of 75 Spanish wetlands are inscribed on this list, covering an area of 304,541 ha. Gallocanta Lake has been included since 1994 [4,5]. In addition, it has been declared a Site of Community Importance (code number ES2430043) according to the Habitat Directive because it contributes significantly to the maintenance of the habitat named "3140 Hard oligo-mesotrophic waters with benthic vegetation of *Chara* spp." [6] of the Natura 2000 Network as defined by the European Environment Agency.

Wetlands are of interest to the scientific community due to their fragility, since they contain migratory and locally endemic species that are threatened by the vulnerability of these to drought events, urban settlements, ecosystem fragmentation, and water imbalances, among other factors [7]. In addition, they contribute to human societies by absorbing 40% of the carbon taken in by carbon sinks, providing protection from severe storms, controlling strong winds, stabilizing coasts, etc. [8].

Gallocanta Lake is a specific type of wetland zone that is currently rare in its biographical area [9], as it is an endorheic brackish lake in the Mediterranean area of the European Western Palaearctic in the supramediterranean continental Iberian region [10]. Furthermore,

it is considered unusual because the water level shows different seasonality and greater fluctuations than other saline lakes in Spain [11,12].

The unique features just mentioned could help to place more emphasis on this lake, which is considered to be the best preserved in Western Europe. For its conservation, a useful tool is remote sensing [13]; it is a simple and optimal way to monitor the uses and coverage of the soil and make the necessary decisions for the management of the territory [9].

The European Space Agency's Copernicus program has several satellites in orbit that provide us with a wealth of information. In our work, we focus on the Sentinel-2 mission, which provides high-resolution multispectral images with which to monitor land and vegetation coverage. The Sentinel-2 mission consists of two satellites: Sentinel-2A launched in June 2015; and Sentinel-2B launched in March 2017. Sentinel-2 was performed to provide knowledge of areas without regular field visits and to provide high-quality data for research. Sentinel mission services are diverse. They include atmospheric surveillance, climate change, marine surveillance, emergencies, security, and ground surveillance [14]. It is this last service we used, because it provides information about vegetation properties, vegetation indicators, dry matter productivity, etc. Sentinel-2 images each consist of twelve individual images which are orthochromatic to their own bandwidths of the electromagnetic spectrum, and they are originally sampled in three different pixel sizes, as shown in Table 1.

**Table 1.** Sentinel-2 satellite bands of the European Copernicus program, operated by the European Space Agency (ESA), indicating the electromagnetic wavelength of the center point of each band and its pixel resolution in meters.

| Band | Wavelength Center (nm) | Pixel Resolution (m) |
|---|---|---|
| Band 1 | 443 nm (dark blue) | 60 m/px |
| Band 2 | 490 nm (light blue) | 10 m/px |
| Band 3 | 560 nm (green) | 10 m/px |
| Band 4 | 665 nm (red) | 10 m/px |
| Band 5 | 705 nm (dark red) | 20 m/px |
| Band 6 | 740 nm (ultra-red) | 20 m/px |
| Band 7 | 783 nm (far red) | 20 m/px |
| Band 8 | 842 nm (near infrared) | 10 m/px |
| Band 8A | 865 nm (near infrared 2) | 20 m/px |
| Band 9 | 940 nm (far infrared) | 60 m/px |
| Band 10 | 1375 nm (short infrared wave) | 60 m/px |
| Band 11 | 1610 nm (short infrared wave) | 20 m/px |
| Band 12 | 2190 nm (short infrared wave 2) | 20 m/px |

Vegetation indexes use the interactions between bands to capture the vegetation present in an image. Sentinel-2's red (band 4) and far-red (bands 5, 6 and 7) bands are designed specifically for vegetation surveys. Several studies have looked into how to obtain the best information about vegetation. Some have even focused on rare habitats and species, such as the works of Charpentier et al. [15] on dwarf eelgrass. Related also to aquatic vegetation detection are the works of Chulafac et al. [16], Fritz et al. [17], Ghirardi et al. [18], and Orth et al. [19], along with Jia et al.'s work [20] on mangroves.

Wetland monitoring is important because of wetlands' roles in improving water quality, flood mitigation, and aquifer recharging. Wetlands support aquatic vegetation, which is essential for some species [21]. In addition, the effect of climate change is especially noticeable in temporary lakes due to their fluctuations in water level and their vulnerability to dry periods. Satellite monitoring of wetlands allows us to appreciate the level of flooding during a flood, and the presence of aquatic vegetation and its phenology in a body of water. This makes it possible to study long time series and observe trends over time [22].

This work aims to evaluate the aquatic vegetation of the Gallocanta saline lake between the years 2015 and 2020 through remote sensing, since it is a powerful tool not only to study the annual distribution and growth patterns influenced by environmental factors such as

drought periods, salinity changes, or frozen waters, which therefore periodically reduce the vegetation in a remarkable way; but also to obtain the most appropriate vegetation index to apply in this study, allowing to know the spatial and temporal heterogeneity in a complementary way to field work. In this way, through the Sentinel-2 Copernicus satellite, we intend to renew the existing information on the vegetation and water quality studies on the area, which has been paralyzed for years, by means of satellite images that provide us with knowledge of the ecological status of the area without losing quality in the research.

## 2. Materials and Methods

### 2.1. Study Site

Gallocanta Lake is an endorreic and saline wetland located between the provinces of Zaragoza and Teruel [23], in the Aragon region, between the counties of Jiloca and Campo de Daroca (40°58′ N/01°30′ W), as shown in Figure 1, including the municipal terms of Gallocanta, Santed, Berrueco and Las Cuerlas (Zaragoza), Bello and Tornos (Teruel). It extends across 1924 hectares of Nature Reserve and 4553 hectares of Peripheral Protection Zone. It has an area of 14.4 km$^2$, with dimensions of 2.8 km wide by 7.7 km long. It has a maximum capacity of 5 hm$^3$ and an average depth of 45–50 cm, although it can reach 3 m in the high-water season [4,24,25].

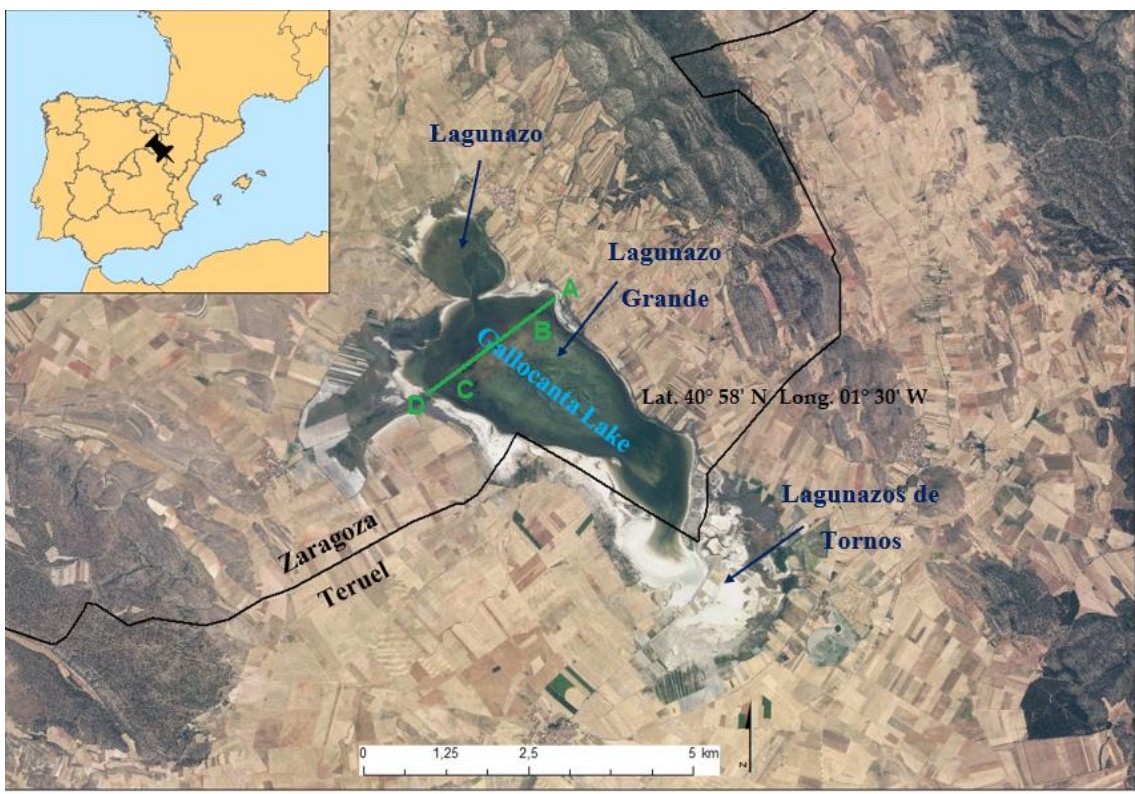

**Figure 1.** Location map of Gallocanta Lake (elaborated from the 2019 image of the National Plan of Air Orthophotograph of the National Geographical Institute of Spain). Geographical coordinates of the center of the lake. Green line A–D indicates vegetation study.

Its basin is between a minimum and maximum altitude of 995 m.a.s.l. (maximum surface lake) and 1085 m.a.s.l., respectively [26]. Due to it is biological, geological, and hydrological uniqueness it is considered one of the most important ecosystems in the Iberian Peninsula and Western Europe. For this reason, this enclave has certain international protection figures such as Ramsar site, Site of European Community Importance, Special Protection Area for Birds (defined in the European Union's Habitats Directive 92/43/EEC), among others regional [4,11].

From a geological point of view, Gallocanta Lake is located at the bottom of an ancient polje, forming part of an endorreic basin with a multifunctional origin, between karstic and structural. Therefore, the lagoon is the sinkhole of the surface runoffs of its basin, accumulating rainwater, which is then lost by evaporation or infiltration. Morphologically (Figure 1), Gallocanta Lake is structured in three areas (north to south): the Lagunazo (lake) and Lagunazo Grande (main lake), the latter being the point of maximum depth, linked by the strait; and the Lagunazos de Tornos (small lakes), to the south, which constitute a flooded wetland area on the occasions that Lagunazo Grande suffers an overflow [11]. A wide description of habitats is presented by Castañeda et al. [25].

The climate present in the lagoon and it is surroundings are semi-arid Mediterranean, common in the steppes of intermediate latitudes. Maximum rainfall occurs between May and June, while minimum rainfall occurs in July, August, January, and February. As for rainfall, there are around 500 mm, with a wide rainfall variation, which produces fluctuations in water level between total drying and almost three meters deep [26]. While the average annual temperature is 10.7 °C, the maximum monthly average recorded in July with 21.1 °C and the minimum in January with 2.9 °C. In addition, it has a great amplitude between the absolute maximum and minimum temperatures, recorded at 39 °C and −21 °C, respectively [27,28].

### 2.2. Dataset and Processing

To carry out the multitemporal analysis to study the evolution of underwater vegetation in the Gallocanta saline lake in different years (2016–2020), we made the process flux presented in Figure 2.

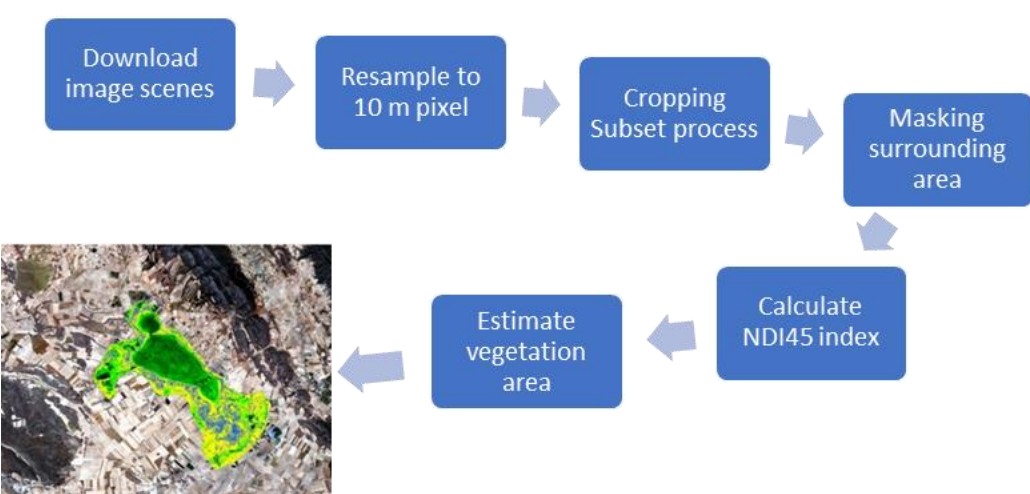

**Figure 2.** Process flux diagram of Sentinel-2 satellite imagery to obtain a thematic map.

The scenes corresponding to the UTM T30XL grid collected by Sentinel-2 of the ESA (European Space Agency) have been downloaded from the Copernicus Open Access Hub portal in https://scihub.copernicus.eu/ (accessed on 26 January 2021) at the MSIL2A processing level already geometrically and atmospherically corrected. Likewise, some images of the NASA portal were also downloaded in https://earthexplorer.usgs.gov (accessed on 15 December 2020), as they are not available on the ESA server at the time of the work. The list of images is presented in Table A1 in the Appendix A. Each image has been processed using Sentinel Application Platform (SNAP, Brockmann Consult Gmbh, Hamburg, Germany) image processing software using the B1-B8A bands, performing a resampling process at 10 m × 10 m pixel; after that, the scene was cropped to the study area (Subset Process), in addition to the display of the image in false color (RGB), masking the surrounding of the protected area and calculating the vegetation index selected.

We used the following indexes, sorted in chronological order of formulation and the formulas applied:

*NDVI* (Normalized Difference Vegetation Index) by Richardson and Wiegand [29]. The more active the chlorophyll of plants, the greater the increase in the level of reflection that occurs in the near-infrared spectrum (*B8*).

$$NDVI = \frac{B8 - B4}{B8 \mp B4} \tag{1}$$

*IPVI* (Infrared Percentage Vegetation Index) by Crippen [30]. Its values range from 0 to 1, where subtracting band 5 from the numerator improves the speed of the calculation.

$$IPVI = \frac{B8}{B8 + B5} \tag{2}$$

*ARVI* (Atmospherically Resistant Vegetation Index) by Kaufman and Tanre [31]. This index takes advantage of the different dispersions of the blue and red band to perform an autocorrect process for the atmospheric effect.

$$ARVI = \frac{B8 - (2 \times B5) + B2}{B8 + (2 \times B5) + B2} \tag{3}$$

*TNDVI* (Transformed Normalized Difference Vegetation Index) by Senseman et al. [32] marks the relationship between the amount of green biomass in a pixel. Its formula is the square root of the *NDVI* with a higher coefficient of determination, and its values will always be positive.

$$TNDVI = \sqrt{\frac{B8 - B4}{B8 + B4} + 0.5} \tag{4}$$

*MCARI* (Modified Chlorophyll Absorption Ratio Index) by Daughtry et al. [33]. The algorithm that has this index is sensitive to the chlorophyll concentrations of the leaf, and to the reflectance of the soil.

$$MCARI = [(B5 - B4) - 0.2 \times (B5 - B3)] \times \frac{B5}{B4} \tag{5}$$

NDI45 (Normalized Difference Index) by Delegido et al. [34]. This index is more linear and with lower saturation at high values than *NDVI*.

$$NDI45 = \frac{B5 - B4}{B5 + B4} \tag{6}$$

The calculation of the indices has been applied on those most appropriate dates (see Table A1) in order to obtain the evolution of underwater vegetation throughout each year. For a more focused approach to the lagoon, a mask has been created that encompasses the lake and the wetland nearest zone. After the creation of the mask, the NDI45 index has been applied to the region of interest, leaving out the study area in RGB.

After that, the area of submerged vegetation was estimated, counting the number of pixels in each class of the index, and the database processed with an Excel spreadsheet (Microsoft Corporation, Redmond, WA, USA) and PAST statistical software [35].

## 3. Results

The maximum area occupied by Gallocanta Lake during the study period was 9.83 km$^2$. This value was reached after the flash flood in mid-April 2018, when the lake went from practically dry to completely full. During April 2020, the maximum area was also reached, but the lake was already covered with water before (not as in 2018).

### 3.1. Vegetation Distribution

According to the work published by Comín et al. [24,36], and from field observation, the existing aquatic vegetation in Gallocanta Lake has been identified. There are three distinct zones in the lake, related to the salinity and depth of the water. The first corresponds to transect A–B (Figure 3a) ubicated in NE of the main body lake (Figure 1), which is a lakeshore zone covered by saline waters (Appendix A, Figure A1).

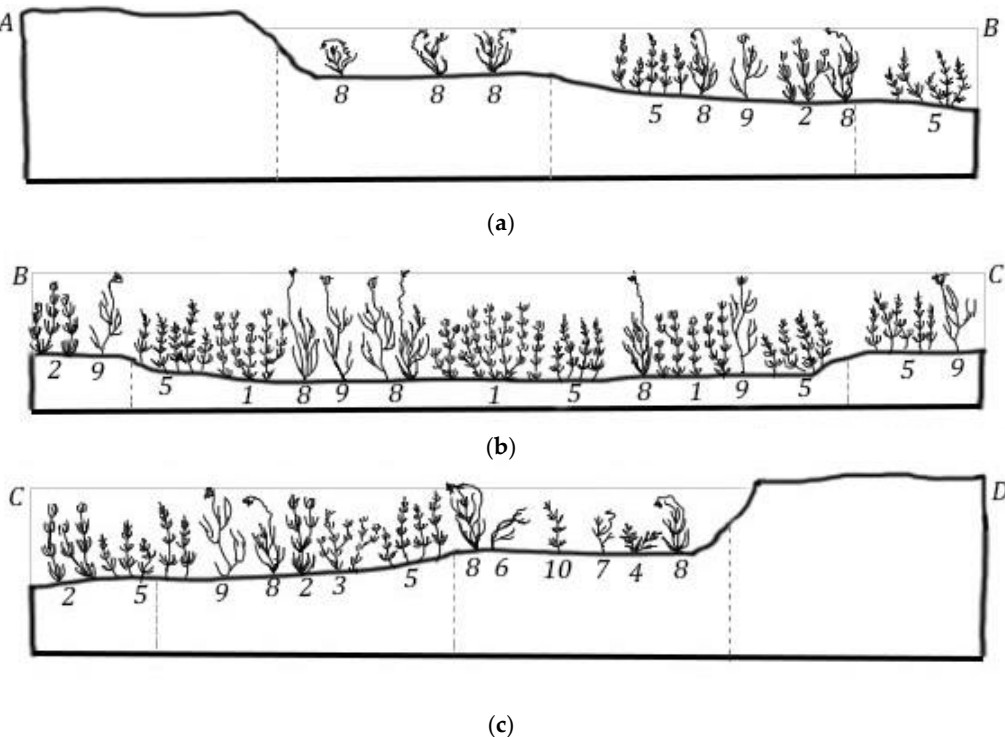

(a)

(b)

(c)

**Figure 3.** (**a**) A–B: Vegetation transect of the saline riparian zone. (**b**) B–C: Vegetation transect of the deeper central zone. (**c**) C–D: Transect of vegetation of the riparian zone influenced by the inflow of water from surface runoff, generally freshwater. Species present [24,36,37]: 1 *Chara galioides*; 2 *Chara foetida*; 3 *Chara hispida*; 4 *Groenlandia densa*; 5 *Lamprothamnium papulosum*; 6 *Potamogeton pectinatus*; 7 *Ruppia cirrhosa*; 8 *Ruppia drepanensis*; 9 *Ruppia maritima*; 10 *Zannichellia palustris*. Drawn by Miriam Ruiz.

The species present are *Chara foetida*, *Lamprothamnium papulosum*, *Ruppia drepanensis*, and *Ruppia maritima*. The central zone of the lake (Figure 3b), which is deeper and also saline, is covered with submerged hydrophytes such as species *Chara galioides*, *C. foetida*, and *L. papulosum* and emergent species *R. drepanensis* and *R. maritima*. Finally, on the banks where there is an inflow of freshwater from surface runoff from the basin (Figure 3c), a vegetation gradient is formed where species *Groenlandia densa*, *Potamogeton pectinatus*, and *Ruppia cirrhosa* are found and *C. foetida*, *Chara hispida*, *L. papulosum*, *P. pectinatus*, *R. drepanensis*, *R. maritima*, and *Zannichellia palustris* are being incorporated. The work carried out periodically by the Basin Authority confirms the current distribution of species in accordance with the previous bibliographic works.

### 3.2. Image Analysis

A total of 105 useful cloud-free images were downloaded between summer 2015 and January 2021 of the scene including Gallocanta lake (Appendix A, Table A1). There are also methods for reconstructing time-series data of vegetation to deal with clod periods [38,39]; however, in our lake, the evolution of vegetation is slow, and we have at least one image per month. The hydroperiod of the lake during this time has been studied in Morales et al. [40] and indicates that during the study time the lake was dry from autumn 2017 to spring 2018. In order to know which vegetation index was more useful for moni-

toring, the calculation was performed for the same spring 2019 image (see Appendix A, Figure A2) in which the vegetated areas in the fields, forests, wetland, and lake could be perfectly observed visually, and thus decide which index was more useful in this case study.

The best performance index was selected comparing the six indices in the image of early spring on 27 March 2019; we evaluated the vegetation area in the RGB false-color image with field data observations (see pictures in Figure A1) and selected the most appropriate index. Other indices do not clearly show the area of the submerged vegetation, giving minor surface compared to NDI45 (Figures 4a and A2). Based on the best difference between vegetated and non-vegetated areas, and the fact that its functional range is from −0.1 to 1, NDI45 [34] was chosen. Values of above 0.1 indicated the presence of vegetation, and we grouped NDI45 values from 0.1 to 0.5 into four classes. This index was originally formulated by the authors experimentally for use with Sentinel-2 but has never been applied to the present for the specific study of aquatic vegetation.

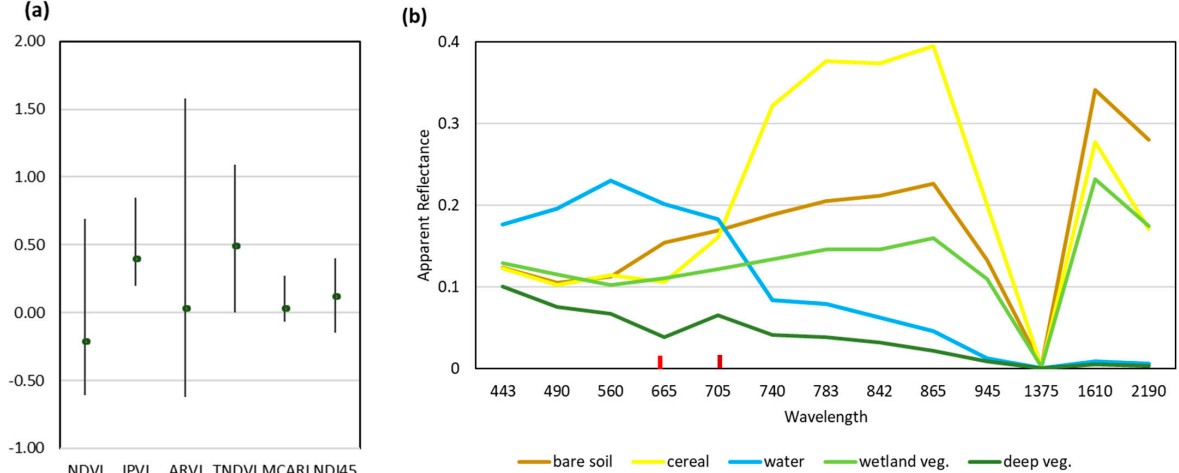

**Figure 4.** (**a**) Comparative values of each index (min, mean, and max values) in the water-table area of the lake. (**b**) Surface reflectance spectrum of the 30 March 2019 image of a point located in bare soil, cereal vegetated land, water, shallow zone aquatic vegetation, and deep zone aquatic vegetation. The red dots indicate the position of image bands 4 (red, 665 nm) and 5 (dark red, 705 nm).

The radiance emitted by vegetation is very different from that emitted by bare soil and water (Figure 4b). Even the spectra emitted by the different types of vegetation observed in an early spring image, when the surrounding fields are growing crops, are different. Considering the applied equation 6 of NDI45, it can be observed that the radiance of water is lower at 705 nm (far red) than at 665 nm (red), so values below zero are typical of water surfaces. In the other cases observed, the radiance value at far red is higher than that at red, and these are positive index values. The radiance values of vegetation are lower the denser it is, so the index value increases as the denominator of equation 6 decreases.

The results of the NDI45 index over the study area and the lake have given values in the range −0.1 to 0.6. The value found in the crops of the area in a good vegetative state were between 0.18 and 0.28. Values below 0 were found in water and in dry soils. In submerged vegetation, values below 0.1 were considered not to indicate the presence of vegetation, while higher values show coverage by submerged vegetation.

The naked eye can see the different images of the dry lagoon, covered with salt crystals (Figure 5a) and the image of lake recently covered with water due to spring storms and snowmelt (Figure 5b). In neither case is there any vegetation in the lagoon, in the first case because of drying and in the second because the flooding was recent, about five days before the image, and the vegetation had not yet developed. As it develops, the hue of the water changes from bluish to light greenish, and then to dark greenish, being easily observable in the natural color images.

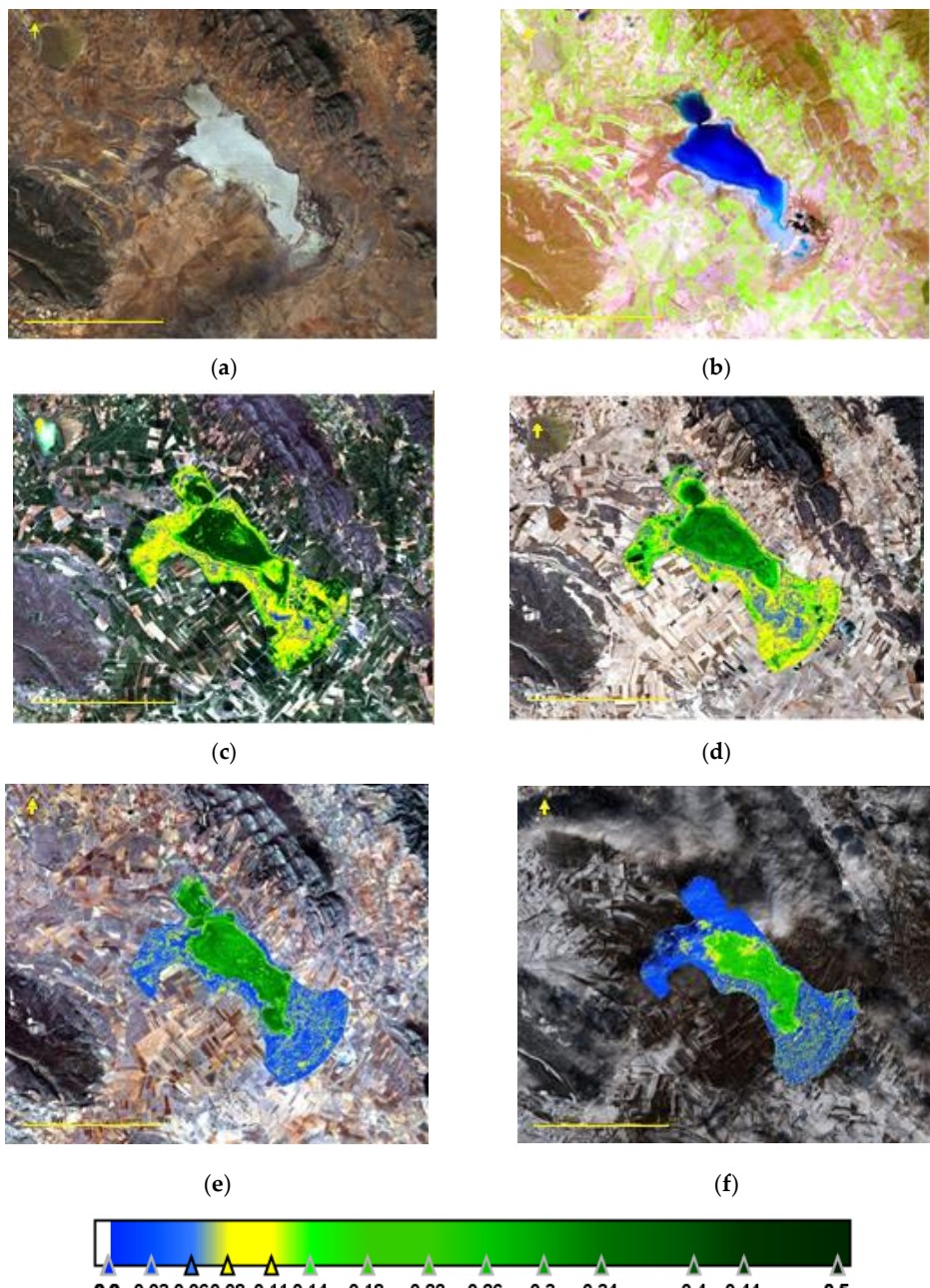

**Figure 5.** Different views of Gallocanta Lake in Sentinel-2 images during the study period. (**a,b**) images without vegetation in RGB natural false color. (**a**) 1 January 2018: Dry lake. (**b**) 19 April 2018: Lake newly inundated by snowmelt and runoff. (**c**) 29 April 2019: vegetation in spring. (**d**) 28 July 2019: vegetation during summer. (**e**) 11 November 2020: vegetation in autumn period. (**f**) 5 January 2021: the lake during snowfall and freezing period. Lower legend vegetation index NDI45. Scale bar 5 km.

### 3.3. Time Series of Vegetated Area

The study of the bottom lake area covered with vegetation over the five-year period using the NDI45 index shows two different cycles. The same color scale has been used for the vegetation index both in the cumulative bar figures and in the thematic maps (see Appendix A, Figures A2–A4), where values higher than 0.1 have been colored in a green hue from light to dark. The first period, between 2015 and 2017 (Figure 6a), presents vegetation in winter and spring. During the summer of 2016 and 2017 (Figure A3), the vegetation practically disappears, leaving only the presence of less than 5% of the surface of the lagoon space. This period ends in the fall of 2017 with the total drying of the lagoon

(Figure 5a), which remains practically without water until the spring of 2018, when we begin the second period of study.

**(a)** NDI45 area 2015-2017

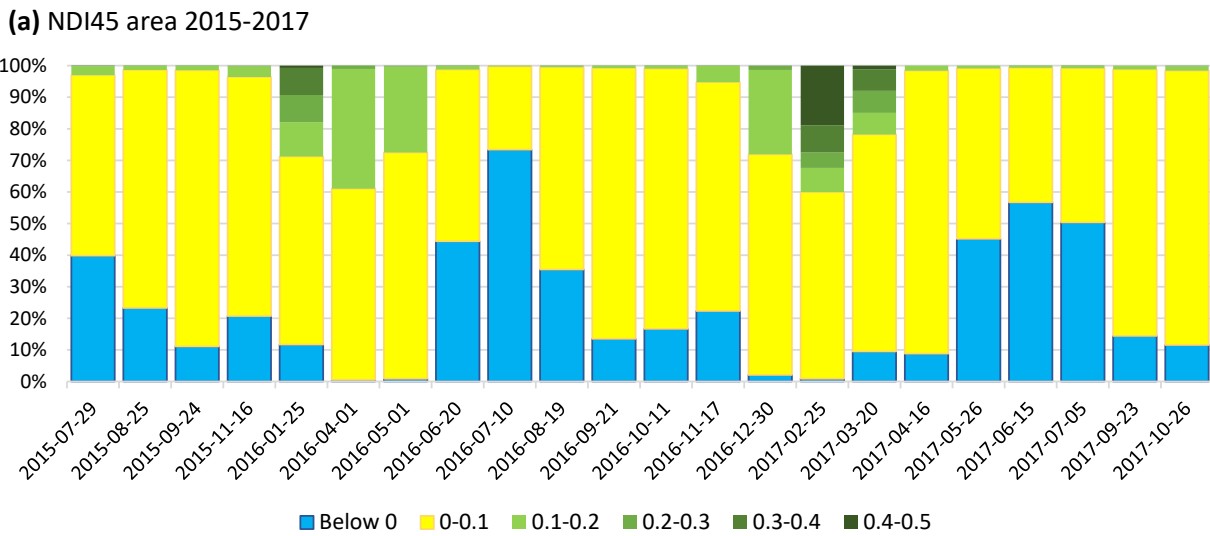

**(b)** NDI45 area 2018 to 2020

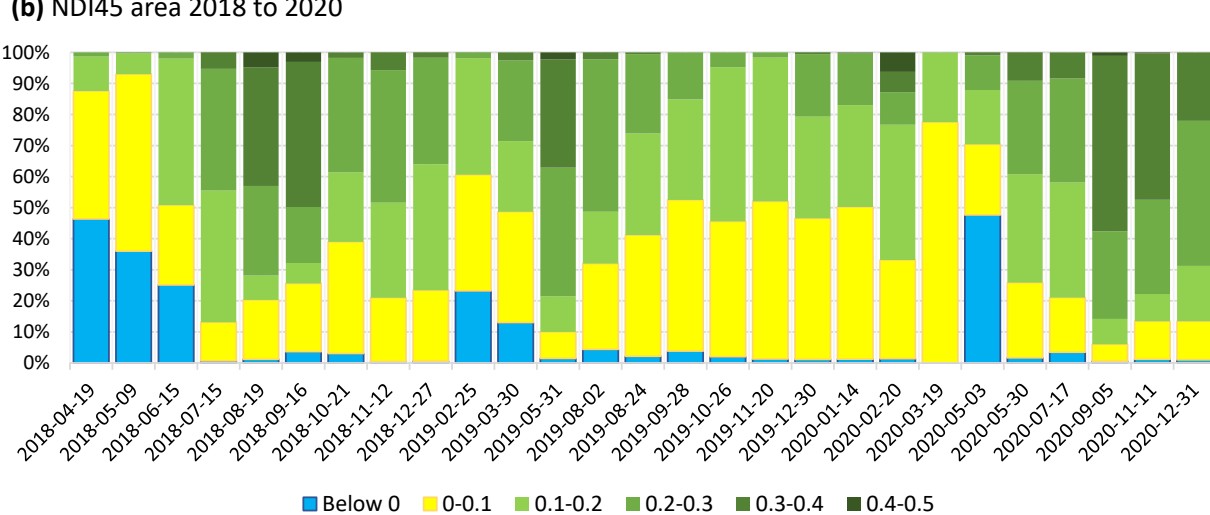

**Figure 6.** Time series in vegetated area in NDI45 classes = 0.1–0.2–0.3–0.3–0.4–0.5. (**a**) Series between summer 2015 and fall 2017, when the lagoon completely dries out. (**b**) Series from spring 2018 (when it floods again) to the end of 2020.

In the second period studied (Figure 6b), after the spring flood of 2019, the lake no longer dried up at any time, so vegetation was maintained throughout the period to a greater or lesser area. The considerable flooding kept the waters at adequate levels and vegetation was observed in the deep zone during the summer of all three years (2018, 2019, and 2020, Figures A4 and A5). The vegetation index showed that aquatic vegetation covers greater than 50% of the flooded area. The largest areas were found in July 2018, May 2019, and September 2020, with values of 90% of the surface occupied by underwater vegetation.

The observation of the images of the study period also showed the impact of storms in the lagoon, and it was possible to observe on some dates the entry of runoff water with suspended materials of earthy color, different from brackish water, for example, the image of 25 April 2020 (Figure 7). During this month, 55 mm of precipitation had already been collected in the area up to this day. On this date, the lake was at its maximum filling level, showing how the natural channels in the northern part of the lake are filled with turbid water. Due to its different composition and density, the mixing is slow and there are areas

of different water quality. After a month, the colorations are no longer observed in the water, probably due to sedimentation of suspended materials and the mixing of fresh and brackish water.

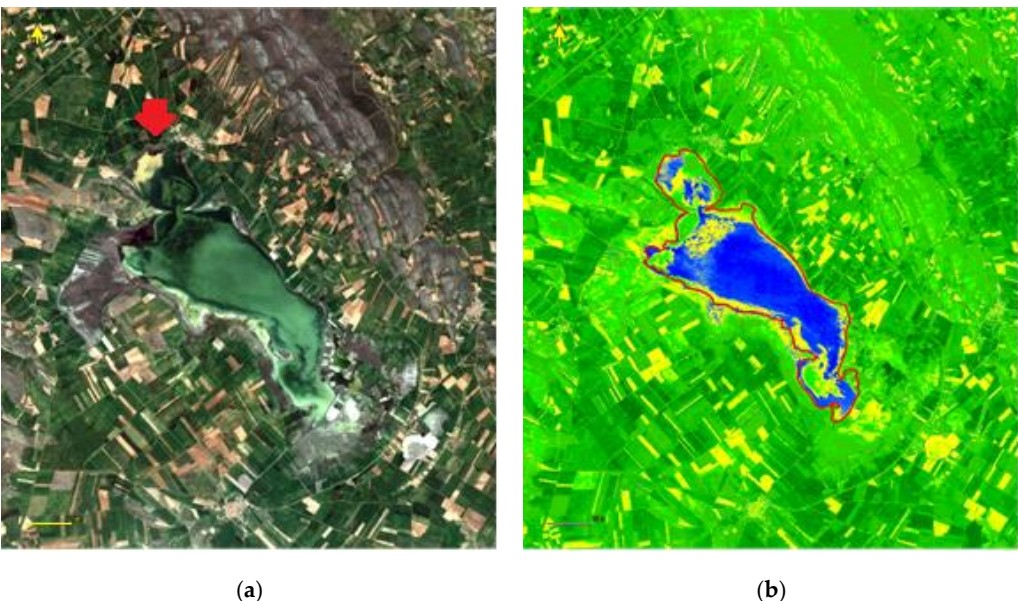

(**a**)            (**b**)

**Figure 7.** (**a**) View of Gallocanta Lake and surrounding area in false natural color on 25 April 2020. After the stormy rains, the entrance of surface runoff water in earthy color is observed from the north, marked with a red arrow. The lake is at maximum level. (**b**) Image of the NDI45 index values. Red line indicates the maximum water table area of the lake studied. Legend of NDI45 values as in Figures 5 and 6. Scale bar 1 km.

The vegetation index identifies as a non-vegetated zone the areas where the water is turbid and has suspended matter, as in the northern zone. The correspondence between the yellow zones of the index (zones without vegetation) and the areas of uncultivated plots is observed, especially in this month, when rainfall was abundant and distributed practically every day.

## 4. Discussion

The importance of aquatic vegetation in the ecosystem is decisive as it supports the food chain and is the first indicator of ecosystem quality [19]. The monitoring of aquatic vegetation by satellite imagery is limited due to the transparency of the water, allowing the observation of vegetation, both on the bottom (hydrophytes) and emerging on the surface (helophytes). At sea, the study of meadows by Sentinel-2 of eelgrass (*Zostera marina*) by Fethers [41] and seagrass (*Posidonia oceanica*) by Traganos and Reinartz [42] has provided good results in terms of vegetation cover maps, but has not allowed taxonomic differentiation of the species present. The importance of knowledge of the spectral signatures of each species is a decisive factor in knowing the individual distribution [43].

In shallow lakes, the presence of both fully submerged and emergent species must be considered, and the influence of the lake edge and the optical properties of the water [43]. In some studies, in deep lakes or in the sea, the study is limited to the area where the bottom is perceived in satellite images. While in the sea depths, up to 50 m is well observed, in lakes, the limit is set to around a 10 m depth. An example is Lake Iseo (Italy), where the work of Ghirardi et al. [18] allows observation of the coverage of *Vallisneria spiralis* and *Najas marina* meadows on the lake margins up to a 10 m depth, distinguishing bare sediment zones and vegetated zones using Sentinel-2 images. However, in shallow lakes, such as Lake Trasimeno (Italy), of about a 4.5 m mean depth, emergent vegetation covers most of the edges forming a macrophyte community dominated by common reed (*Phragmites australis*)

together with other emergent species such as *Typha angustifolia* and submerged species such as the genera *Potamogeton*, *Myriophyllum*, *Chara*, and *Ceratophyllum* [44]. Observation by means of aerial images made it possible to map the distribution of plant communities. Similar work was also carried out on the lakeshore of Lake Balaton (Hungary) by Stratoulias et al. [45], who, using Sentinel-2 imagery, also assessed lakeshore vegetation and showed its usefulness in monitoring small-scale habitats.

The monitoring of underwater vegetation in the shallow lake of Gallocanta is a difficult task to perform by conventional surveys in the lake itself, given its extension and the depth of the water body, which does not facilitate navigation. It is also not easy to evaluate the presence of plants systematically because the lake is considered a special protection area for birds and there are access limitations regarding the presence of certain species in the area, such as flamingos. For this reason, the Basin Authority carries out only occasional censuses. It should be noted that there has been only four published studies on aquatic vegetation in the last forty years, due to the difficulty in carrying them out. The lack of studies is also influenced by the fact that the lake is in an unpopulated area, lacks impacts except for those derived from agriculture dedicated to the cultivation of cereals, and has a good ecological status, so it has no conservation problems. The results obtained in this study are consistent with the last sampling carried out by the administration [37], in which they obtained a coverage of 10% due to submerged plants and the presence of filamentous algae. The total estimate would be at 20%, similar to that obtained with the satellite image from a date very close to the sampling day.

The absence of further field data prevents us from distinguishing vegetation types between emergent macrophytes, submerged macrophytes, and filamentous algae. The study of the different classes of the index suggests that values between 0.1 and 0.2 correspond to filamentous algae, while values higher than 0.2 correspond to macrophytes. Further studies should expand on these details of species coverage in specific areas in order to improve the interpretation of the index results.

Studies of underwater vegetation in other lakes show the presence of species similar to those of Gallocanta Lake, despite its high salinity. Coincidences occur in areas under freshwater influence, with species such as *Zannichellia palustris* and *Potamogeton pectinatus*, which have also occurred on the margins of Lake Stanberg (Germany) [17], whose monitoring was also carried out with Sentinel-2 images.

As for species typical of saline waters, the genus *Ruppia* is typical of them, as pointed out by the work of Verhoeven [46] in several European saline lakes, and especially in temporary lakes that dry out once a year, as noted by Brock [47]. However, *Lamprothamnium papulosum* is a species distributed in saline lakes of all types, including tidal lakes, having been cited in saline lakes in Greece [48] or in the studies of Charpentier et al. in the coastal lagoon of Vaccares (France) [15]. The coincidence of the species *L. papulosum*, *R. maritima* and *R. drepanensis* between Gallocanta Lake and the saline lake of Fuente de Piedra (Málaga, Spain) has been highlighted by the works of Conde-Alvarez et al. [49] and also with the saline lake of Chiprana (Zaragoza, Spain) where they also occur [50]. *R. maritima*, also known as widgeon-grass, is a species that is included as "endangered" in the Catalogue of Threatened Species of Aragon, approved by Decree 49/1995, 28 March, of the Government of Aragon.

Finally, it should be noted that the study using remote-sensing images provides good general results of the aquatic vegetation cover, although it requires support from field work to improve the identification of the taxa present and their spectral response in the image. Future studies should be oriented towards in situ radiometric measurements and their comparison with the results obtained by reflectance.

## 5. Conclusions

The degree of coverage of the water body by submerged and emergent aquatic vegetation has been evaluated from the images obtained from the Sentinel-2 satellite of Gallocanta Lake. The study is limited by the frequency of the images and by the turbidity of the water when surface runoff occurs due to rainfall. Water turbidity impedes the view of submerged

vegetation communities. The results obtained show that vegetation growth is limited by the seasonality of the lake. When the lake dries out, salt concentration occurs and the vegetation decays and disappears at the time of drying. In the period studied, the lake had dried up from autumn 2017 to spring 2018, coinciding with a low-rainfall winter. From the filling in April 2018, the lake did not lower in water level again; therefore, the presence of aquatic vegetation became continuous until the end of the study in December 2020. During this period, the vegetation presented the maximum coverage during the summer, reaching up to 90% of the surface, while the lowest values were observed during the winter.

**Author Contributions:** Conceptualization: M.R., J.S. and S.M.; methodology: S.M. and J.S.; formal analysis: M.R., J.S. and S.M.; investigation: M.R., J.S. and S.M.; data curation: J.S.; writing and editing: J.S.; review: M.R. and S.M.; supervision: J.S. All authors have read and agreed to the published version of the manuscript.

**Funding:** This research received no external funding.

**Institutional Review Board Statement:** Not applicable.

**Data Availability Statement:** Sentinel-2 imagery were downloaded from the Scihub Copernicus site. Meteorological data are available from the AEMET Spanish web service.

**Acknowledgments:** The authors express their gratitude to R. Muñoz for providing the pictures of vegetation (Figure A1).

**Conflicts of Interest:** The authors declare no conflict of interest.

## Appendix A

**Table A1.** List of Universal Resource Identifiers of images freely downloaded from the cloud. * Files processed for the NDI45 vegetation index.

1. S2A_MSIL1C_20150729T110026_N0204_R094_T30TXL_20150729T110648 *
2. S2A_MSIL1C_20150825T105046_N0204_R051_T30TXL_20150825T105942 *
3. S2A_MSIL1C_20150924T105046_N0204_R051_T30TXL_20150924T105656 *
4. S2A_MSIL1C_20151116T110322_N0204_R094_T30TXL_20151116T110736 *
5. S2A_OPER_MSI_L1C_TL_SGS__20160112T162938_A002908_T30TXL_N02
6. S2A_OPER_MSI_L1C_TL_SGS__20160125T164509_A003094_T30TXL_N02 *
7. S2A_OPER_MSI_L1C_TL_SGS__20160204T150456_A003237_T30TXL_N02
8. S2A_OPER_MSI_L1C_TL_SGS__20160401T180421_A004052_T30TXL_N02 *
9. S2A_OPER_MSI_L1C_TL_SGS__20160414T162250_A004238_T30TXL_N02
10. S2A_OPER_MSI_L1C_TL_SGS__20160501T144220_A004481_T30TXL_N02 *
11. S2A_OPER_MSI_L1C_TL_SGS__20160620T161042_A005196_T30TXL_N02 *
12. S2A_OPER_MSI_L1C_TL_SGS__20160710T161148_A005482_T30TXL_N02 *
13. S2A_OPER_MSI_L1C_TL_SGS__20160819T161132_A006054_T30TXL_N02 *
14. S2A_OPER_MSI_L1C_TL_SGS__20160921T162916_A006526_T30TXL_N02 *
15. S2A_OPER_MSI_L1C_TL_SGS__20161011T162433_A006812_T30TXL_N02 *
16. S2A_OPER_MSI_L1C_TL_SGS__20161028T161846_A007055_T30TXL_N02
17. S2A_OPER_MSI_L1C_TL_SGS__20161117T180146_A007341_T30TXL_N02 *
18. S2A_MSIL1C_20161207T105432_N0204_R051_T30TXL_20161207T105428
19. S2A_MSIL1C_20161230T110442_N0204_R094_T30TXL_20161230T110441 *
20. S2A_MSIL1C_20170225T105021_N0204_R051_T30TXL_20170225T105020 *
21. S2A_MSIL1C_20170320T105731_N0204_R094_T30TXL_20170320T110651 *
22. S2A_MSIL1C_20170416T105031_N0204_R051_T30TXL_20170416T105601 *
23. S2A_MSIL1C_20170526T105031_N0205_R051_T30TXL_20170526T105518 *
24. S2A_MSIL1C_20170605T105031_N0205_R051_T30TXL_20170605T105303
25. S2A_MSIL1C_20170608T105651_N0205_R094_T30TXL_20170608T110453
26. S2A_MSIL1C_20170615T105031_N0205_R051_T30TXL_20170615T105505 *
27. S2A_MSIL1C_20170705T105031_N0205_R051_T30TXL_20170705T105605 *
28. S2A_MSIL1C_20170923T105021_N0205_R051_T30TXL_20170923T105717 *
29. S2A_MSIL1C_20171026T110131_N0206_R094_T30TXL_20171026T144303 *
30. S2A_MSIL1C_20171122T105341_N0206_R051_T30TXL_20171122T131313
31. S2A_MSIL1C_20180101T105441_N0206_R051_T30TXL_20180101T131633
32. S2B_MSIL1C_20180419T105619_N0206_R094_T30TXL_20180419T121006 *
33. S2B_MSIL1C_20180509T105619_N0206_R094_T30TXL_20180509T113046 *
34. S2B_MSIL1C_20180615T105029_N0206_R051_T30TXL_20180615T130739 *
35. S2B_MSIL1C_20180715T105029_N0206_R051_T30TXL_20180715T144536 *

**Table A1.** *Cont.*

36. S2A_MSIL1C_20180819T105031_N0206_R051_T30TXL_20180819T130740 *
37. S2B_MSIL1C_20180916T105649_N0206_R094_T30TXL_20180916T131145 *
38. S2A_MSIL1C_20181021T110101_N0206_R094_T30TXL_20181021T112408 *
39. S2B_MSIL1C_20181112T105259_N0207_R051_T30TXL_20181112T125827 *
40. S2B_MSIL1C_20181212T105439_N0207_R051_T30TXL_20181212T124901
41. S2A_MSIL1C_20181227T105441_N0207_R051_T30TXL_20181227T112349 *
42. S2A_MSIL1C_20181230T110441_N0207_R094_T30TXL_20181230T113426
43. S2B_MSIL1C_20190213T110149_N0207_R094_T30TXL_20190213T162521
44. S2A_MSIL1C_20190225T105021_N0207_R051_T30TXL_20190225T125616 *
45. S2A_MSIL1C_20190330T105631_N0207_R094_T30TXL_20190330T130620 *
46. S2B_MSIL1C_20190531T105039_N0207_R051_T30TXL_20190531T130936 *
47. S2B_MSIL1C_20190802T105629_N0208_R094_T30TXL_20190802T131125 *
48. S2A_MSIL1C_20190804T105031_N0208_R051_T30TXL_20190804T130903
49. S2B_MSIL1C_20190809T105039_N0208_R051_T30TXL_20190809T125039
50. S2A_MSIL1C_20190814T105031_N0208_R051_T30TXL_20190814T112158
51. S2A_MSIL1C_20190817T105621_N0208_R094_T30TXL_20190817T132102
52. S2B_MSIL1C_20190822T105629_N0208_R094_T30TXL_20190822T131655
53. S2A_MSIL1C_20190824T105031_N0208_R051_T30TXL_20190824T125731 *
54. S2A_MSIL1C_20190903T105031_N0208_R051_T30TXL_20190903T131033
55. S2A_MSIL1C_20190906T105621_N0208_R094_T30TXL_20190906T130522
56. S2B_MSIL1C_20190908T105029_N0208_R051_T30TXL_20190908T125351
57. S2A_MSIL1C_20190926T105811_N0208_R094_T30TXL_20190926T130550
58. S2B_MSIL1C_20190928T105029_N0208_R051_T30TXL_20190928T130836 *
59. S2B_MSIL1C_20191001T105849_N0208_R094_T30TXL_20191001T131419
60. S2A_MSIL1C_20191003T105031_N0208_R051_T30TXL_20191003T111605
61. S2B_MSIL1C_20191008T105029_N0208_R051_T30TXL_20191008T125041
62. S2B_MSIL1C_20191011T105959_N0208_R094_T30TXL_20191011T121547
63. S2A_MSIL1C_20191026T110141_N0208_R094_T30TXL_20191026T113511 *
64. S2B_MSIL1C_20191120T110249_N0208_R094_T30TXL_20191121T135932 *
65. S2B_MSIL1C_20191207T105329_N0208_R051_T30TXL_20191207T111410
66. S2B_MSIL1C_20191210T110339_N0208_R094_T30TXL_20191210T113810
67. S2A_MSIL1C_20191225T110451_N0208_R094_T30TXL_20191225T113446
68. S2B_MSIL1C_20191230T110349_N0208_R094_T30TXL_20191230T113841 *
69. S2A_MSIL1C_20200111T105421_N0208_R051_T30TXL_20200111T112046
70. S2A_MSIL1C_20200114T110411_N0208_R094_T30TXL_20200114T113817 *
71. S2A_MSIL1C_20200203T110241_N0208_R094_T30TXL_20200203T113113
72. S2B_MSIL1C_20200205T105129_N0209_R051_T30TXL_20200207T121337
73. S2A_MSIL1C_20200220T105051_N0209_R051_T30TXL_20200319T122308 *
74. S2A_MSIL1C_20200223T110041_N0209_R094_T30TXL_20200223T114220
75. S2A_MSIL1C_20200314T105821_N0209_R094_T30TXL_20200314T130531
76. S2B_MSIL1C_20200319T105649_N0209_R094_T30TXL_20200319T132209 *
77. S2A_MSIL1C_20200403T105621_N0209_R094_T30TXL_20200403T130550
78. S2B_MSIL1C_20200425T104619_N0209_R051_T30TXL_20200425T130643
79. S2A_MSIL1C_20200503T105621_N0209_R094_T30TXL_20200503T130649 *
80. S2A_MSIL1C_20200520T105031_N0209_R051_T30TXL_20200520T125512
81. S2A_MSIL1C_20200523T105631_N0209_R094_T30TXL_20200523T130313
82. S2A_MSIL1C_20200530T105031_N0209_R051_T30TXL_20200530T125407 *
83. S2A_MSIL1C_20200622T105631_N0209_R094_T30TXL_20200622T133306
84. S2A_MSIL1C_20200629T105031_N0209_R051_T30TXL_20200629T125331
85. S2B_MSIL1C_20200704T104619_N0209_R051_T30TXL_20200704T125053
86. S2B_MSIL1C_20200707T105619_N0209_R094_T30TXL_20200707T130404
87. S2B_MSIL1C_20200717T105619_N0209_R094_T30TXL_20200717T130432 *
88. S2A_MSIL1C_20200719T105031_N0209_R051_T30TXL_20200719T125648
89. S2A_MSIL1C_20200729T105031_N0209_R051_T30TXL_20200729T130448
90. S2A_MSIL1C_20200801T105631_N0209_R094_T30TXL_20200801T113256
91. S2A_MSIL1C_20200808T105031_N0209_R051_T30TXL_20200808T111510
92. S2B_MSIL1C_20200826T105619_N0209_R094_T30TXL_20200826T131208
93. S2A_MSIL1C_20200831T105621_N0209_R094_T30TXL_20200831T131423
94. S2B_MSIL1C_20200902T104629_N0209_R051_T30TXL_20200902T130348
95. S2B_MSIL1C_20200905T105619_N0209_R094_T30TXL_20200905T121813 *
96. S2A_MSIL1C_20200910T105631_N0209_R094_T30TXL_20200910T131324
97. S2B_MSIL1C_20200912T104629_N0209_R051_T30TXL_20200912T132431
98. S2B_MSIL1C_20200915T105639_N0209_R094_T30TXL_20200915T131429
99. S2A_MSIL1C_20200927T105031_N0209_R051_T30TXL_20200927T130634
100. S2B_MSIL1C_20201111T105259_N0209_R051_T30TXL_20201111T130651 *
101. S2B_MSIL1C_20201121T105349_N0209_R051_T30TXL_20201121T115417
102. S2A_MSIL1C_20201209T110441_N0209_R094_T30TXL_20201209T131140
103. S2A_MSIL1C_20201226T105451_N0209_R051_T30TXL_20201226T130209
104. S2B_MSIL1C_20201231T105349_N0209_R051_T30TXL_20201231T120402 *
105. S2A_MSIL1C_20210105T105441_N0209_R051_T30TXL_20210105T130129 *

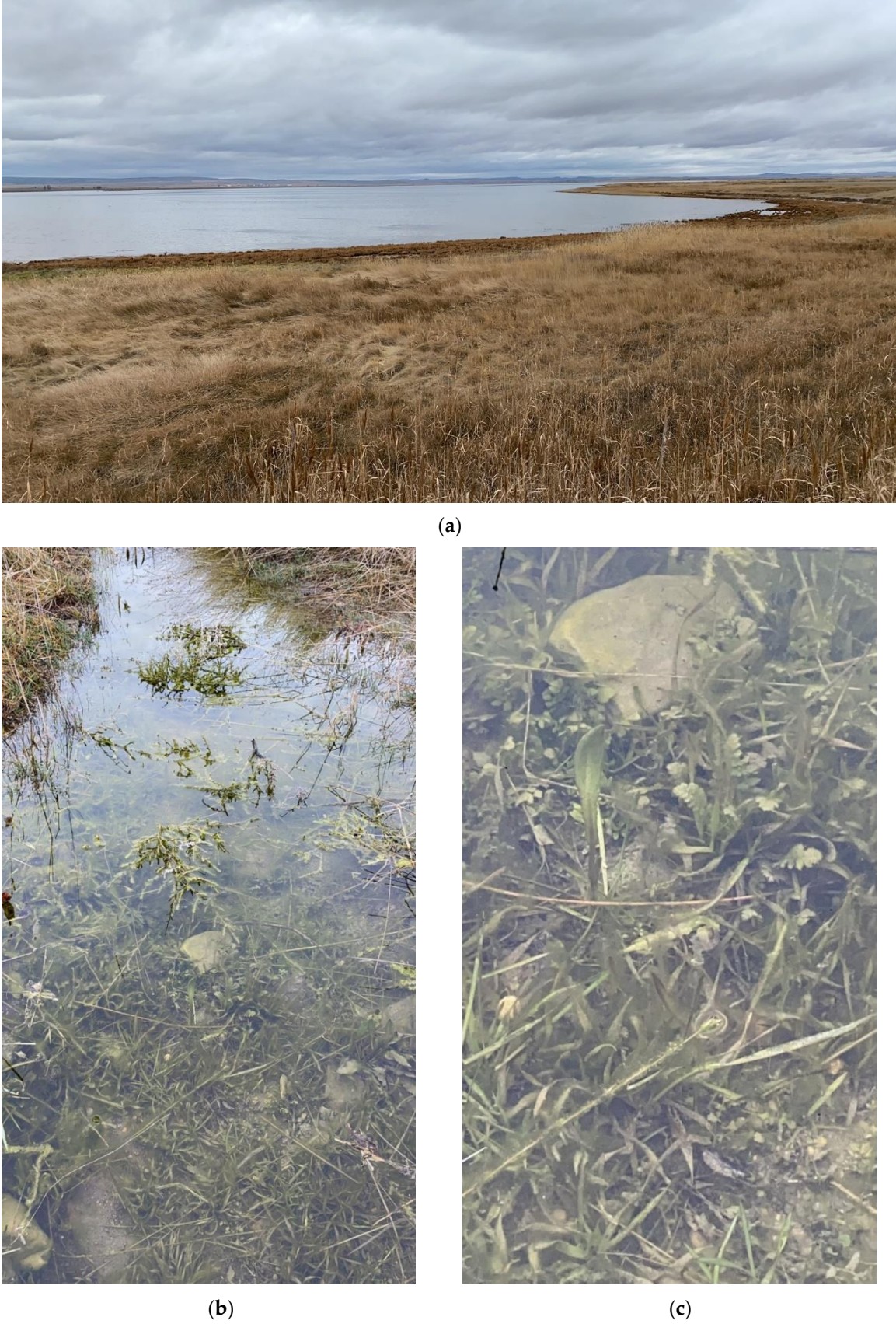

**Figure A1.** Pictures of vegetation in Gallocanta Lake shore: (**a**) General view from Northeast area in autumn (transect zone A–B in Figure 1). (**b**,**c**) Detail of submerged aquatic plants.

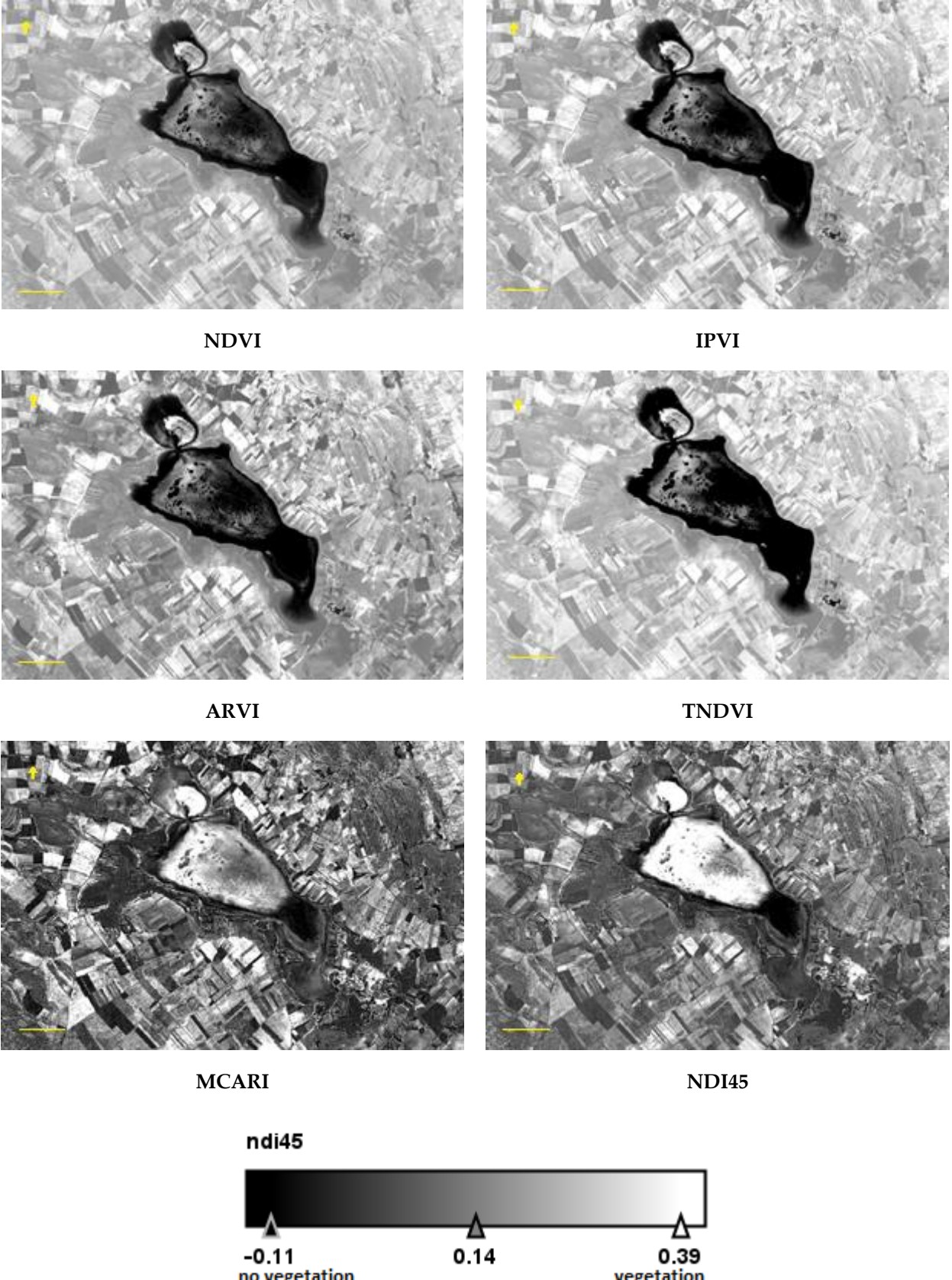

**Figure A2.** Comparison between indices: image of 27 March 2019. Scale bar: 1 km.

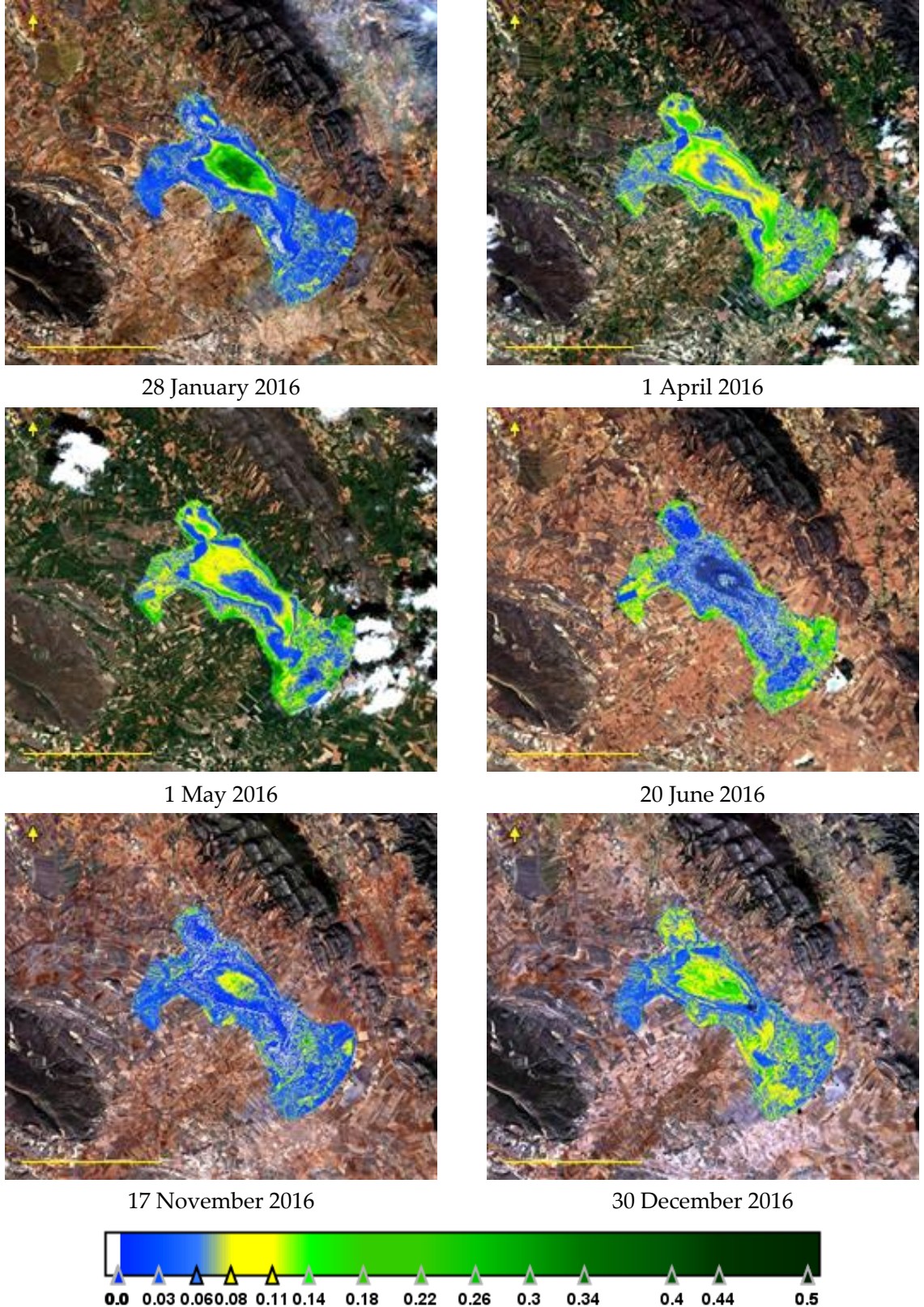

28 January 2016

1 April 2016

1 May 2016

20 June 2016

17 November 2016

30 December 2016

0.0  0.03 0.06 0.08 0.11 0.14  0.18  0.22  0.26  0.3  0.34    0.4  0.44    0.5

**Figure A3.** NDI45 evolution during 2016. Scale bar: 5 km.

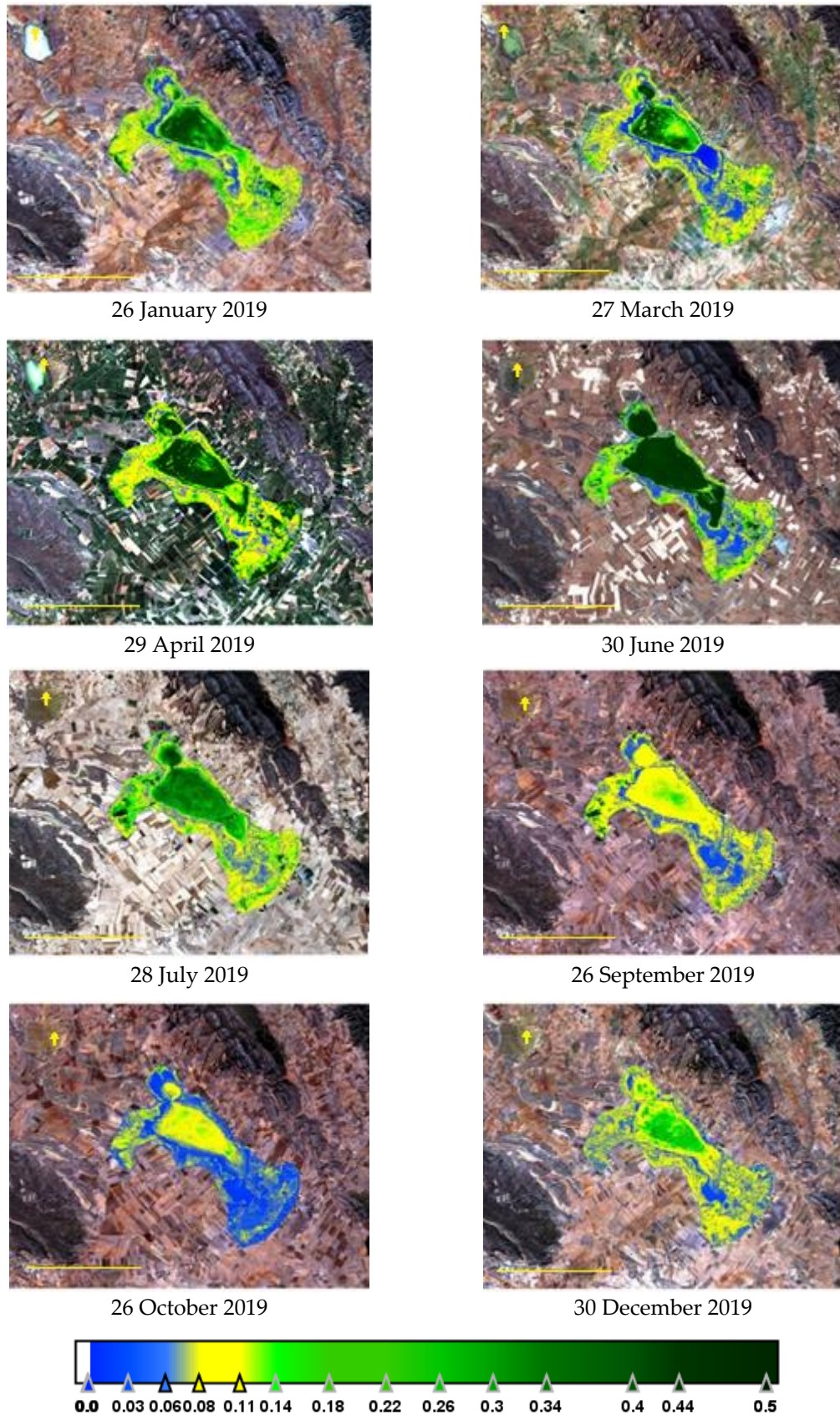

**Figure A4.** NDI45 evolution during 2019. Scale bar: 5 km.

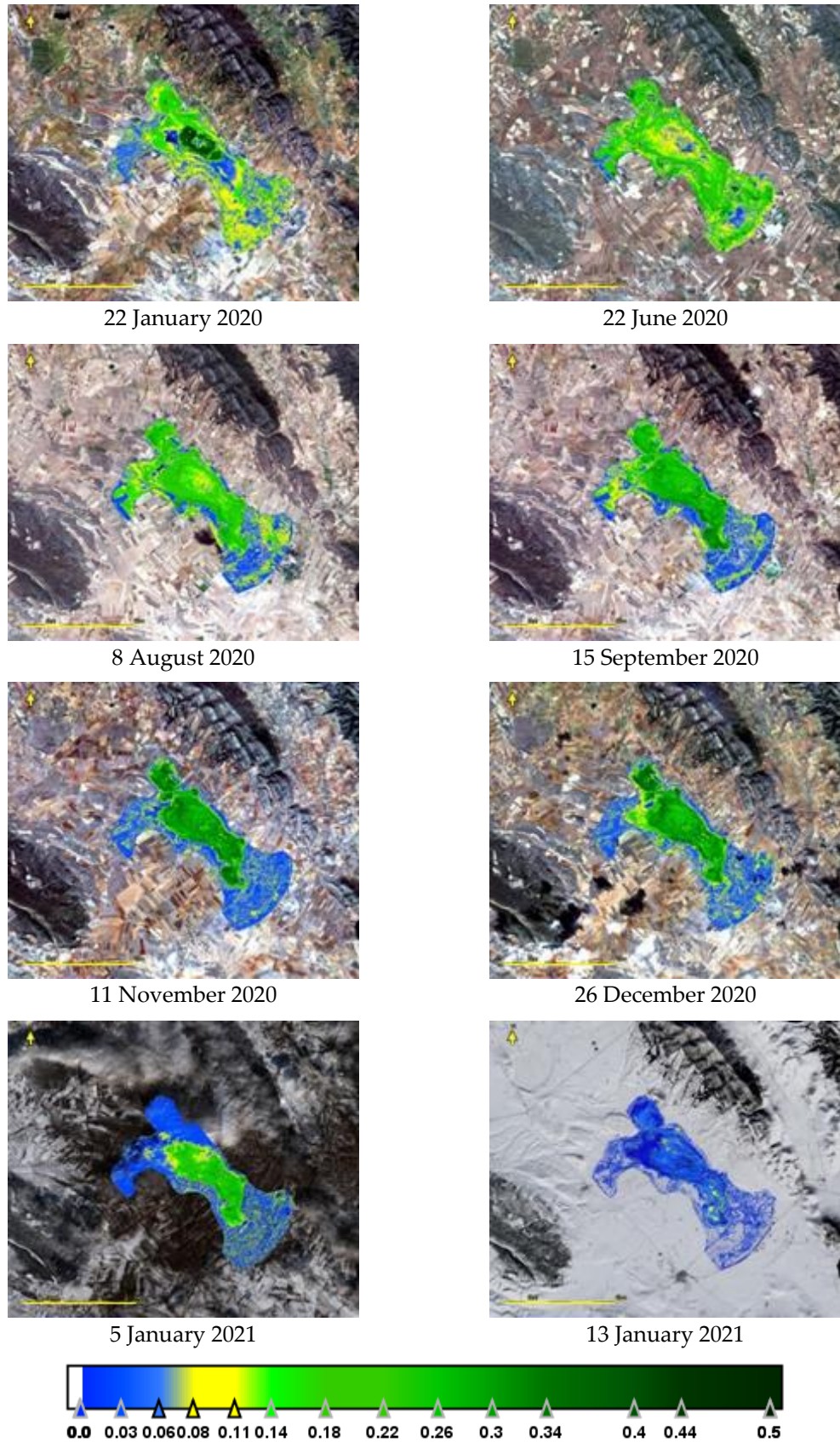

**Figure A5.** NDI45 evolution over a year: January 2020 to January 2021. Scale bar: 5 km.

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
