# Peer review of "Monitoring Subaquatic Vegetation Using Sentinel-2 Imagery in Gallocanta Lake (Aragón, Spain)"

_2673-4834, doi:10.3390/earth3010022_

Round 1

Reviewer 1 Report

This manuscript tries to use vegetation indices from the Sentinel-2 imageries to monitor the subaquatic vegetation in Gallocanta Lake, Spain. Six different typical vegetation indices are used, and the best one is found. There is a certain meaning for this manuscript, however, I think many issues should be improved before it can be published. Here are some comments for sharing.

  • The structure of the Introduction section is disordered and confusing. It would be better to separate the introduce of the remote sensing method and the Gallocanta Lake. The remote sensing tool is introduced in Line 38-40, but the next paragraph turns to introduce the Gallocanta Lake, and the next two paragraphs introduce the remote sensing again. It is quite confused to understand. The authors should reorganize the Introduction to be logical.
  • Line 72-88, the six vegetation index should not be detailed introduced in this section, but should be in the Section of the method. Here can just be a brief introduce of different vegetation indices.
  • For Section 2-4, i.e., Materials and Methods, Results, and Discussion, sub-sections and sub-titles should be added to help the reader to understand the manuscript. I find it is so hard to know how the authors do their experiments and what results they obtain in the current version. If the sub-sections are added, so we can find the contents I want directly, and we can easily get logic of the experiments.
  • The information of all the applied Sentinel-2 imageries should be included in the Materials section, such as the acquisition date.
  • Please clarify how you assess the performance of different vegetation indices, and add the content in the Method section.
  • I cannot find the tight connection between the method and the conclusions obtained. It seems that the authors want to use the vegetation index to monitor the subaquatic vegetation, however, I find that most of the experiments use the images in RGB natural false color.
  • Line 341, it is declared that the study is limited by the availability of cloud-free images. Actually, there are many vegetation index reconstruction methods to deal with this issue [1-4], which can support the long-term continuous study in the future. This should be discussed in the Discussion section.

[1] Zeng L, Wardlow B D, Hu S, et al. A Novel Strategy to Reconstruct NDVI Time-Series with High Temporal Resolution from MODIS Multi-Temporal Composite Products[J]. Remote Sensing, 2021, 13(7): 1397.

[2] Chen, J.M., Deng, F., Chen, M., 2006. Locally adjusted cubic-spline capping for reconstructing seasonal trajectories of a satellite-derived surface parameter. IEEE Trans. Geosci. Remote Sens. 44, 2230–2237

[3] Chu Dong, Shen Huanfeng, et. al., Long time-series NDVI reconstruction in cloud-prone regions via spatio-temporal tensor completion. Remote Sensing of Environment. 264. 2021.

[4] Chen, J., Jonsson, ¨ P., Tamura, M., Gu, Z., Matsushita, B., Eklundh, L., 2004. A simple method for reconstructing a high-quality NDVI time-series data set based on the Savitzky-Golay filter. Remote Sens. Environ. 91, 332–344.

Author Response

This manuscript tries to use vegetation indices from the Sentinel-2 imageries to monitor the subaquatic vegetation in Gallocanta Lake, Spain. Six different typical vegetation indices are used, and the best one is found. There is a certain meaning for this manuscript, however, I think many issues should be improved before it can be published. Here are some comments for sharing.

 Thank you very much for your valuable comments, that contribute to improve the manuscript.

  • The structure of the Introduction section is disordered and confusing. It would be better to separate the introduce of the remote sensing method and the Gallocanta Lake. The remote sensing tool is introduced in Line 38-40, but the next paragraph turns to introduce the Gallocanta Lake, and the next two paragraphs introduce the remote sensing again. It is quite confused to understand. The authors should reorganize the Introduction to be logical.

Paragraphs are moved as you propose and presented in a better logical order.

  • Line 72-88, the six vegetation index should not be detailed introduced in this section, but should be in the Section of the method. Here can just be a brief introduce of different vegetation indices.

Description of vegetation indices are moved to methods section, and in the introduction are only described.

  • For Section 2-4, i.e., Materials and Methods, Results, and Discussion, sub-sections and sub-titles should be added to help the reader to understand the manuscript. I find it is so hard to know how the authors do their experiments and what results they obtain in the current version. If the sub-sections are added, so we can find the contents I want directly, and we can easily get logic of the experiments.

Subsections are added to the manuscript.

  • The information of all the applied Sentinel-2 imageries should be included in the Materials section, such as the acquisition date.

Due to the great number of images, we add a table with the dates in the annex section, as Table A1.

  • Please clarify how you assess the performance of different vegetation indices, and add the content in the Method section.

We add a paragraph with this content in the end of methods section. We explain that the best performance index was selected comparing the six indices in the image of early spring of 2019 in March 27; we evaluate the vegetation area in the RGB false color image and selected which was the most appropriate index. Other indices do not show clearly the area of the submerged vegetation, giving minor surface that NDI45 (Figure A1). Values of above 0.1 indicated the presence of vegetation, and we grouped in four classes from 0.1 to 0.5 NDI45 values.

  • I cannot find the tight connection between the method and the conclusions obtained. It seems that the authors want to use the vegetation index to monitor the subaquatic vegetation, however, I find that most of the experiments use the images in RGB natural false color.

This appreciation is not correct. We only use the RGB natural false color to select the most appropriate index from the six considered, comparing the area of vegetation and the different indices in the image of March 2019. We present this in the methods section now. Also in results we use the RGB image to observe the drought to flood period (figure 4a,b) and the frozen period (figure 4f). We do not process images when the lake is dry.

After that, we monitor the vegetation counting the pixels in each class of the index NDI45 (from 0.1 to 0.5) and add this text now in the methods. The results of the counting are presented in figure 5.

  • Line 341, it is declared that the study is limited by the availability of cloud-free images. Actually, there are many vegetation index reconstruction methods to deal with this issue [1-4], which can support the long-term continuous study in the future. This should be discussed in the Discussion section.

 Thank you very much for your references. We have incorporated 3 & 4 in position 38 and 39 and discussed this point about the clod-free images and the reconstruction for time series with these methods. In our lake, the evolution of the vegetation is slow, and we can monitor the surface coverage easily.

[38] Chu Dong, Shen Huanfeng, et. al., Long time-series NDVI reconstruction in cloud-prone regions via spatio-temporal tensor completion. Remote Sensing of Environment. 264. 2021.

[39] Chen, J., Jonsson, ¨ P., Tamura, M., Gu, Z., Matsushita, B., Eklundh, L., 2004. A simple method for reconstructing a high-quality NDVI time-series data set based on the Savitzky-Golay filter. Remote Sens. Environ. 91, 332–344.

Reviewer 2 Report

Dear Authors,

The study describes employing approach for monitoring  subaquatic vegetation using Sentinel-2 imagery in a karstic lake  of central Spain  (Gallocanta Lake). To accomplish this goal in this study were tested how remote sensing data can be allows the monitoring  of aquatic vegetation cover in a shallow lake from the different spectral response of the water when the vegetation grows on the bottom and reaches the surface. 
Overall, the paper touches a topical subject in the context of Remote Sensing applied to freshwater habitat classification. However, there are few issues to be addressed before the manuscript can be suitable for publication. Please following my comments and suggestions in the attached PDF file.
My review response of this paper is accepted with major revision. 
The English in the present manuscript require minor improvement, please carefully proof-read spell check to eliminate it.

Good Luck!

Author Response

The study describes employing approach for monitoring  subaquatic vegetation using Sentinel-2 imagery in a karstic lake  of central Spain  (Gallocanta Lake). To accomplish this goal in this study were tested how remote sensing data can be allows the monitoring  of aquatic vegetation cover in a shallow lake from the different spectral response of the water when the vegetation grows on the bottom and reaches the surface. 
Overall, the paper touches a topical subject in the context of Remote Sensing applied to freshwater habitat classification. However, there are few issues to be addressed before the manuscript can be suitable for publication. Please following my comments and suggestions in the attached PDF file.
My review response of this paper is accepted with major revision. 
The English in the present manuscript require minor improvement, please carefully proof-read spell check to eliminate it.

Thank you very much for your valuable comments. We have incorporated your suggestions and modified according to your review and the other two reviewers.

Reviewer 3 Report

The present study suggest a way to monitor subaquatic vegetation using Sentinel-2 imagery. The manuscript in its present form needs some important adjustments before going ahead. After performing it I will certainly recommend it for the publication on Earth. Despite the case of study is interesting some revision must be done... Here they are some advices that can improve the paper.

Introduction

The introduction is too short i suggest you to better discribe the background of subaquatic vegetation monitoring taking also into account the effect of climate change; here they are some papers to cite:

  • Kaplan, G., and U. Avdan. "MAPPING AND MONITORING WETLANDS USING SENTINEL-2 SATELLITE IMAGERY." ISPRS Annals of Photogrammetry, Remote Sensing & Spatial Information Sciences 4 (2017).
  • https://doi.org/10.3390/cli9030047
  • https://doi.org/10.3390/w11030563

Secondly I suggest you to introduce after the introduction section, a new one called study area (you can use the part written between lines 104-136 and also the first image in the introduction) The first image is a cartography but in this present form, it doesn't provide any important and, as well as, necessary information please add in it or in the caption section below (Datum, Projection, Representation scale, Nominal scale and Title).

Materials and Methods

I suggest you to divide these sections in two parts. In materials describe the dataset adopted Sentinel-2 which level? The software adopted like QGIS and so on... In methods the workflow of the processing performed and also insert the part that it is already present into the paper.

Results

Please better describe the statistics performed and in Figure 3 replace Digital Level with Surface Reflectance or better describe what is Digital Level it is not clear and can induce readers in errors.

In all images if you decide that are maps follow the above mentioned advices and introduce a legend please.

The results are enough fine in this present form, you only have to support this statement in line 344 "When evaporation and salt concentration occur, the vegetation decays and disappears at the time of drying." in your manuscript by analysis or citation of other workw related to evapotraspiration and salt and vegetation in your area of study. Differently I suggest you to remove it. 

Author Response

The present study suggest a way to monitor subaquatic vegetation using Sentinel-2 imagery. The manuscript in its present form needs some important adjustments before going ahead. After performing it I will certainly recommend it for the publication on Earth. Despite the case of study is interesting some revision must be done... Here they are some advices that can improve the paper.

Introduction

The introduction is too short i suggest you to better describe the background of subaquatic vegetation monitoring taking also into account the effect of climate change; here they are some papers to cite:

Thank you very much for your proposes. We have incorporated the first and second references in number 21 and 22 (lines 77-83). Note that the third one yet is included in the introduction (reference 18) and discussion as a reference for some of the taxon that are also present in Gallocanta lake as Vallisneria and Najas.

  • Kaplan, G., and U. Avdan. "MAPPING AND MONITORING WETLANDS USING SENTINEL-2 SATELLITE IMAGERY." ISPRS Annals of Photogrammetry, Remote Sensing & Spatial Information Sciences 4 (2017).
  • https://doi.org/10.3390/cli9030047
  • https://doi.org/10.3390/w11030563

Secondly I suggest you to introduce after the introduction section, a new one called study area (you can use the part written between lines 104-136 and also the first image in the introduction) The first image is a cartography but in this present form, it doesn't provide any important and, as well as, necessary information please add in it or in the caption section below (Datum, Projection, Representation scale, Nominal scale and Title).

Other reviewers also propose the same, and we have created the new section in Methods and incorporated the paragraphs indicated and made a new wording.

Materials and Methods

I suggest you to divide these sections in two parts. In materials describe the dataset adopted Sentinel-2 which level? The software adopted like QGIS and so on... In methods the workflow of the processing performed and also insert the part that it is already present into the paper.

We reworded the section and incorporate two sections, the first one for study site and the second one for dataset and processing. A workflow has drawn as figure 2.

Results

Please better describe the statistics performed and in Figure 3 replace Digital Level with Surface Reflectance or better describe what is Digital Level it is not clear and can induce readers in errors.

Digital level is replaced by Surface reflectance, it was a mistake.

In all images if you decide that are maps follow the above mentioned advices and introduce a legend please.

We include the geographical information in thematic maps, as proposed, indicating the north arrow and a segment scale, and explained in the foot of figures.

The results are enough fine in this present form, you only have to support this statement in line 344 “When evaporation and salt concentration occur, the vegetation decays and disappears at the time of drying.” In your manuscript by analysis or citation of other workw related to evapotraspiration and salt and vegetation in your area of study. Differently I suggest you to remove it. 

We observed in satellite images the decay of vegetation when the lake lost the level of water. This result is presented in the figures 5a and 6.  

Round 2

Reviewer 1 Report

The authors have made efforts to revise the manuscript, but I think there are still two issues that need further modification.

  • The comparison of results from different vegetation indexes is described in Section 2.2 (Line 176-181). I think these contents should belong to the first of the Section of results, and the corresponding Figure A2 (I think it should be Figure A2 rather than Figure A1 in Line 179) should be moved to the main text of the manuscript. The Section of data and method should only describe how you compare and evaluate the different results.
  • I still find it hard to compare the different results in Figure A2, where is the ground truth data. Please provide the legend of the different colors.
  • The title of section 3.3 is questionable.

Author Response

The authors have made efforts to revise the manuscript, but I think there are still two issues that need further modification.

Thank you very much again for your dedication and interest in the manuscript.

  • The comparison of results from different vegetation indexes is described in Section 2.2 (Line 176-181). I think these contents should belong to the first of the Section of results, and the corresponding Figure A2 (I think it should be Figure A2 rather than Figure A1 in Line 179) should be moved to the main text of the manuscript. The Section of data and method should only describe how you compare and evaluate the different results.

We moved the text from methods to the section 3.2 of results and merged the text, now in lines 219-228. We consider not to move the table A2 from appendix to the main text because is big and the manuscript has many figures. But we make a small graph bar with the numeric results of each index in figure 4a.

  • I still find it hard to compare the different results in Figure A2, where is the ground truth data. Please provide the legend of the different colors.

We make a numerical estimation area in the six indices and presented in figure 4a. We provided the graphical legend scale for vegetation index in figure A2.

  • The title of section 3.3 is questionable.

We have changed to “Time series of vegetated area”.

Reviewer 2 Report

Dear Authors,

I have read carefully the manuscript reviewed, the authors have carefully considered the comments and tried the best to address every one of them. My response is accepted in present form and ready for the next steps of publish processing.

All the best

Author Response

Thank you for your comments. We have make some minos changes as requested by reviewer 1.

Reviewer 3 Report

The authors well aimed all the suggestions proposed and now the manuscript is clear and well presented in each part. Therefore, I certainly reccomend the pubbliciation of the paper in this present form.

Author Response

Thank you for your dedication to review. We also have made minor changes as requested by reviewer 1.